# Radiotherapy, Chemotherapy and Immunotherapy—Current Practice and Future Perspectives for Recurrent/Metastatic Oral Cavity Squamous Cell Carcinoma

**DOI:** 10.3390/diagnostics13010099

**Published:** 2022-12-29

**Authors:** Cecília Melo-Alvim, Maria Eduarda Neves, Jorge Leitão Santos, André N. Abrunhosa-Branquinho, Tiago Barroso, Luís Costa, Leonor Ribeiro

**Affiliations:** 1Department of Medical Oncology, Centro Hospitalar Universitário Lisboa Norte, 1649-028 Lisboa, Portugal; 2Department of Radiotherapy, Centro Hospitalar Universitário Lisboa Norte, 1649-028 Lisboa, Portugal; 3Luís Costa Lab, Instituto de Medicina Molecular–João Lobo Antunes, Faculdade de Medicina de Lisboa, 1649-028 Lisboa, Portugal

**Keywords:** squamous cell carcinoma, oral cavity, treatment option, recurrent/metastatic disease

## Abstract

Oral squamous cell carcinoma is the most common malignant epithelial neoplasm affecting the oral cavity. While surgical resection is the cornerstone of a multimodal curative approach, some tumors are deemed recurrent or metastatic (R/M) and often not suitable for curative surgery. This mainly occurs due to the extent of lesions or when surgery is expected to result in poor functional outcomes. Amongst the main non-surgical therapeutic options for oral squamous cell carcinoma are radiotherapy, chemotherapy, molecular targeted agents, and immunotherapy. Depending on the disease setting, these therapeutic approaches can be used isolated or in combination, with distinct efficacy and side effects. All these factors must be considered for treatment decisions within a multidisciplinary approach. The present article reviews the evidence regarding the treatment of patients with R/M oral squamous cell carcinoma. The main goal is to provide an overview of available treatment options and address future therapeutic perspectives.

## 1. Introduction

Malignant tumors of the lip and oral cavity are the 16th most common tumors in humans [1]. Oral cancers account for the majority of squamous cell carcinomas of the head and neck (SCCHN). Their prevalence is variable, with rates as low as 5% of all cancers in the United States to as high as 30–45% in India [1]. The survival 5 years after diagnosis is around 40–50%, with the global burden of oral cancer documented to increase between 1990 and 2017 [1,2]. Approximately 2.4 deaths per 100,000 and 64.2 disability-adjusted life years (DALYs) lost per 100,000 people were reported in 2017. This burden of disease is higher in men [2]. Although the quality of care for these patients has been increasing in most countries, global disparities still exist worldwide due to uneven access to healthcare between and within countries [2,3]. Several regional and global exogenous risk factors have been described for oral cancer, where tobacco, either smoked or chewed, is one of the main factors. The prevalence of SCCHN is 5 to 8.4 times higher in tobacco users compared to the general population [4]. Alcohol, another risk factor, is directly harmful to the DNA and acts synergistically with tobacco as a carcinogen [4]. Despite not being considered for treatment selection in international guidelines, human papillomavirus (HPV)16/18 is an acknowledged carcinogen, which will potentially be used as a prognostic and predictive marker in the future [5,6,7]. Genetic studies have uncovered a great variety of alterations associated with these tumors, such as aberrations in the fragile histidine triad gene (probably related to tobacco smoke), co-amplification of EIF3E and RECQL4 (related to alcohol consumption), and PIK3CA mutations (with increased prevalence in HPV-associated disease). Tumor mutations or epigenetic changes may lead to abnormal cell signaling and proliferation, malignant transformation, and evasion from the host immune response [4]. Patient symptoms are usually associated with invasion of adjacent structures (and may include limitation of tongue mobility, trismus, dysphagia, stridor) or treatment side effects. Treatment for localized disease consists of a combination of surgery, radiotherapy, and systemic therapy (either conventional cytotoxic chemotherapy or targeted therapy with cetuximab). Even when it is not operable, localized SCCHN is potentially curable with systemic therapy and radiotherapy. This range of therapeutic options warrants the need for a multidisciplinary approach to these patients, encompassing medical oncologists, radiation oncologists, and surgeons, among other health professionals, within a shared decision process. While the side effects of local (radiotherapy and/or surgery) and systemic therapy have an impact on patients’ quality of life (QoL) in the short term [8,9,10], long-term survivors of localized disease have a good the QoL at 2 to 10 years after end of treatment [11,12], although inferior to controls with no history of cancer [13]. Active smoking at the time of diagnosis and the need to use feeding tube in the long term predict low QoL [11]. In recurrent/metastatic (R/M) disease setting, patients’ QoL decreases rapidly as a result of treatments or disease progression, leading to impairment in activities of daily living and poor health-related QoL, and placing a high burden on formal and informal caregivers [14]. This highlights the need to optimize the management of this patient population. The present article reviews the state of the art of therapeutic options for R/M SCCHN, including radiotherapy, chemotherapy, targeted therapy, and immunotherapy.

## 2. Radiotherapy

The standard of care for locoregional recurrent SCCHN is surgical resection, followed by adjuvant chemotherapy and/or radiotherapy, if indicated according to pathologic risk factors [6].

For patients deemed medically unfit for surgery or for whom surgery would result in adverse functional outcomes, radical radiotherapy, with or without chemotherapy, is an option. The University of Chicago first demonstrated the feasibility of this approach in 1996, after reviewing previously irradiated head and neck cancer patients enrolled in four phase I/II trials [15]. The investigators reported a 2-year overall survival (OS) of ≈20%, which increased to 35% among patients receiving over 58 Gray (Gy). The long-term analysis of this study confirmed initial results, with the reirradiation dose remaining an independent prognostic factor [16]. However, the late toxicity of this treatment was significant, with 19 of 115 patients dying from treatment-related intercurrences, including 5 from carotid artery blowout.

Another series of 169 reirradiated head and neck patients from Institut Gustave Roussy yielded similar results, with a 2-year OS of 20% [17]. However, five patients died from carotid blowout, and late grade 3 toxicities were frequent. The RTOG 96-10 trial investigated a twice-daily radiation schedule with delivery of 60 Gy of radiation to previously irradiated head and neck patients, with comparable results: 1-year OS of 48%, but 7% of treatment-related deaths [18]. In this study, the time between each radiation course and reirradiation for a second primary tumor instead of for local recurrence significantly correlated with survival. This difference in prognosis between second and recurrent primary tumors was already suggested in a series of 100 patients reirradiated with external beam plus brachytherapy [19]. If indicated, radiotherapy and/or chemotherapy should be considered even for patients submitted to surgery for recurrent local disease. A randomized trial from Institut Gustave-Roussy assigned 130 patients with macroscopically resected recurrent head and neck cancers (18% of which were oral primary tumors) to adjuvant chemoradiation or observation [20]. Although adjuvant treatment did not provide a survival advantage (2-year OS 40–50%), locoregional control was significantly improved in the adjuvant treatment group, as well as deaths related to local recurrence. Late toxicity was noticeably higher in the adjuvant group, including grade 3–4 sclerosis, trismus, and osteoradionecrosis, which affected as much as 39% of surviving patients at 2 years. The development of more precise, dose-intensive radiation regimens for other anatomical sites spurred the interest on stereotactic body radiotherapy (SBRT) for locally recurrent head and neck tumors.

A phase I dose-escalation trial including seven patients with recurrent oral cavity cancers explored up to 44 Gy in five fractions [21]. Although no grade 3 toxicities were observed, only a modest interval of 4 months until disease progression was reported. Another study used a median of 30 Gy in five fractions at reirradiation reported more promising results, although 15% of patients died of bleeding due to carotid blowout [22]. Interestingly, the authors found that this outcome was only observed in patients whose tumors completely encased the carotid artery.

The finding that cetuximab conferred a survival benefit versus radiotherapy alone in naïve head and neck patients unfit for platinum-based chemotherapy led investigators from the University of Pittsburgh to conduct a matched case–control study of SBRT in recurrent disease setting [23,24]. Patients treated with concomitant cetuximab had a median OS of ~24 months and no grade 4–5 toxicities. A phase II trial also exploring SBRT (36 Gy in six fractions) and concurrent cetuximab showed a more modest OS of 11.8 months, with one treatment-related death [25]. These results were confirmed in another phase II trial including 29% of patients with recurrent oral cavity cancer, which reported an OS of 10 months and only 6% of grade 3 or higher late toxicities [26].

Radiotherapy with curative intent may not be feasible in head and neck tumors for a variety of reasons, including simultaneous relapse with local recurrence and distant metastases, patients’ unfitness for aggressive treatments (i.e., low performance status (PS)), and/or patient choice. In these cases, radiotherapy may play a relevant role as palliative treatment of these tumors, including those of the oral cavity. The goal of palliative radiotherapy for the primary tumor is local relief with some degree of local control, particularly in patients without indication for curative treatment [27].

Well-established palliative schemes considering patients’ PS and estimated survival are available. The ideal patient candidate should have Eastern Cooperative Oncology Group (ECOG) PS 0–2 (or 3 in selected cases) and an estimated survival over 1–3 months. Patients should tolerate the positioning of the procedure, and its potential benefit (both from a clinical and patient perspective) should outweigh the discomfort it causes. The well-known QUAD-SHOT regimen includes at least one radiotherapy cycle that corresponds to two twice-daily treatments of 3.7 Gy on two consecutive days. An interval of 2–4 weeks is required between each cycle to assess response, toxicity, and the need for replanning. The time interval between the new treatment planning and execution of a new cycle should not exceed 3 days, in order to allow for treatment on consecutive days. Corry et al. assessed 30 eligible cases with the QUAD-SHOT regimen, showing that 43% of patients had oral cavity carcinoma [28]. Although only 16 patients completed three cycles, 16 patients had an objective response (2 cases with complete response) and 7 had stable disease. The median OS was 5.7 months, and the median progression-free survival (PFS) was 3.1 months. Toxicity was assessed in 27 patients, 14 of whom experienced grade 1 radiodermitis, 9 experienced grade 1 mucositis, and 3 experienced grade 2 mucositis. Other schemes can be applied following the recommendations adapted from Grewal et al. Besides PS, the radiation oncologist should consider the previous history of irradiation in overlapping fields [29,30].

In patients with low estimated survival (up to 4 months) unfit for other cancer treatments, the main goal is comfort and symptom relief. Ideally, the duration of the complete radiotherapy scheme should be no longer than two weeks. Possible schemes include QUAD-SHOT, 20 Gy in five fractions (one fraction per day), and 28 Gy in three fractions (in days 0, 7, and 21) [30,31,32,33]. For patients with an estimated survival between 4 and 12 months, QUAD SHOT with or without chemotherapy is an option, as well as 20 Gy in 5 fractions (4 Gy daily), 30 Gy in 5 fractions twice per week, or 40 Gy in 10 fractions twice per week [30,31,32,33]. For patients with an estimated survival over 12 months, more aggressive treatment can be considered. For patients with no indication for other treatments, a hypofractionated regimen with 50 Gy in 16 fractions (3.125 Gy/day) or 52.5 Gy in 15 fractions (3.5 Gy/day) is recommended [30].

The fact that palliative radiation has a primary goal of symptom relief should not hinder the fact that it can also contribute to local control and even survival. The QUAD-SHOT study reported over 50% of objective responses, with a median OS of 5.7 months, which is promising given that those were patients not amenable to curative therapy. “Christie scheme” (3.125 Gy per fraction) reported an OS of 40% at 1 year and a median survival time of 17 months. Even the more modest “0–7–21” regimen showed a median 6-month OS of 51% with a 39% PFS within the irradiated volume [31].

Brachytherapy also has a place in the treatment of head and neck cancer, with oral cavity tumors being the best candidates for this approach. However, its efficacy evidence comes mainly from retrospective studies. New guidelines have been recently published by GEC-ESTRO ACROP for the use of brachytherapy as a reirradiation option in inoperable patients, according to which this approach allows for adequate coverage without serious toxicities, such as bone invasion or fistula. Brachytherapy can also be used as a boost after external beam radiation therapy [32,33,34].

Some currently ongoing trials may uncover new directions for the treatment of R/M SCCHN, as the combination of radiotherapy with immunotherapy, which is currently a hot topic for investigation. The pillar concept for this combination is a synergistic effect, by which neoantigens produced during radiotherapy treatment and the immunotherapy agents (i.e., immune checkpoint inhibitors) promote immunological synapses in order to intensify the host immune system against the cancer cells. This can happen near the area of irradiation, but also over distant metastasis (known as abscopal effect). This combination can be useful for locally advanced disease and metastatic disease, especially if limited oligoprogression is observed while on isolated immunotherapy, promoting the so-called “turning cold tumor to hot tumor” effect [35]. Although more established for the combination of immune therapies, the immune effect produced by radiotherapy can also play a role in this setting.

The concept is feasible but there are still impeding questions beyond the scope of the paper that are under investigation: What are the best immunotherapy agents for the combination? Which is the ideal biomarker(s)? Ideal timings for the introduction of each therapy? What are the appropriate RT technique, prescription dose, and treatment volumes of interest to enhance the immunological effect?

Ongoing trials without results for a plethora of cancer diseases and settings are being conducted with very few SCCHN cancers, especially for oral cavity carcinomas. Although not exclusively in oral cavity tumors, the KEYSTROKE/RTOG 3507 phase II trial (NCT03546582) is comparing SBRT alone versus SBRT in combination with pembrolizumab in locoregionally recurrent or second primary head and neck cancers. rEA3191 is another phase II trial seeking to compare reirradiation with pembrolizumab versus re-irradiation plus paclitaxel versus pembrolizumab alone in locally recurrent or second primary SCCHN in a previously irradiated field [36,37].

Potential combinations with novel radioenhancers, such as nanoparticles, are also being investigated. NBTXR3 is a hafnium oxide crystalline nanoparticle compound that is injected directly into tumors to enhance the absorption of ionizing radiation, resulting in increased tumor cell death without adding toxicity to adjacent normal tissues. This approach has shown promising results in soft tissue sarcoma [38,39]. It is being investigated across multiple tumor types and different settings, including inoperable locoregional recurrent SCCHN. A phase II trial with two cohorts of patients with inoperable locoregional recurrent SCCHN is currently active and recruiting patients (NCT04834349). In the cohort I, the aim of the study is to estimate the PFS and early clinical benefit of NBTXR3 activated by SBRT reirradiation with concurrent pembrolizumab. Cohort II aims to assess the safety profile and estimate early clinical benefit of NBXTR3 activated by dose reduction of intensity-modulated radiation therapy (IMRT) or intensity-modulated proton therapy (IMPT) reirradiation with concurrent pembrolizumab in patients with locoregional recurrent disease not eligible for SBRT [40].

Keypoints of Radiotherapy section:Patients with R/M oral cavity SCCHN cancers impose a challenge since the impossibility of surgery hampers clinical outcomes.Reirradiation with external beam radiotherapy can be offered, but patient selection is important to decide the treatment intent (curative vs. palliative) because OS is limited and severe cumulated toxicities are increased (e.g., carotid blowout, trismus, and osteonecrosis).In curative reirradiation, the 2-year OS is around 20%, but treatment-related events can reach up to 7–15% of cases. Most data are based on IMRT techniques and very few with stereotactic treatments.There are a wide variety of palliative RT schemes to confer best comfort and symptom relief, with the QUAD-SHOT regimen being the most known.Robust data for brachytherapy techniques are lacking, and ongoing trials are being conducted to search the benefit RT combinations with novel agents, such as immunotherapy or nanoparticles.

## 3. Chemotherapy and Molecular Target Agents

In initial treatment approach to patients with R/M SCCHN, systemic therapy is chosen based on exposure to previous therapies, time since completion of definitive treatment, anatomic distribution and burden of the current disease, and toxicity of previous systemic treatments [5]. Other prognostic factors to consider when choosing the treatment approach are the patient’s PS and comorbidities, tumor-programmed death molecule-1 (PD- L1) expression status, and symptoms related to disease burden [5].

Among chemotherapy options, platinum agents, such as cisplatin and carboplatin, are used both as single agents and the backbone for most combination regimens in head and neck tumors [41,42,43]. Although there is little supporting evidence from head-to-head studies, carboplatin is often considered less effective than cisplatin, being preferred in some settings due to its low potential for neurotoxicity or nephrotoxicity and despite being with myelosuppression [41]. Besides platinum agents, other cytotoxic options include taxanes, which have shown response rates of 20–40% in phase II trials and may be an option in monotherapy for patients with renal dysfunction and contraindication for cisplatin [44]. Other options with some reported benefit include methotrexate and fluorouracil, although these agents have lower response rates compared to others and no survival impact [45]. Cetuximab, an epidermal growth factor receptor (EGFR)-targeted monoclonal antibody, has shown activity alone and in combination with chemotherapy [46,47]. Conversely, there is no established role for the anti-EGFR panitumumab or for the anti-vascular endothelial growth factor (VEGF) bevacizumab in this setting [48,49].

For patients with advanced head and neck cancer previously untreated or who completed treatment more than 6 months before disease progression, the new standard of care was established in the KEYNOTE-048 trial, which will be further addressed in the Immunotherapy Section of this article [50]. For patients with contraindication to immunotherapy, treatment options include doublet cytotoxic chemotherapy regimens with or without concurrent cetuximab [47]. Doublet cytotoxic chemotherapy has been shown to increase the objective response rate (ORR) compared with single-agent chemotherapy, although with no survival benefit [51]. Patients only eligible for single-agent therapies (mostly due to poor PS, comorbidities, or previous lines of therapy) may be considered for treatment with taxanes, methotrexate, fluorouracil, or cetuximab [51]. The EXTREME randomized phase III trial included a population of 442 patients predominantly linked to tobacco and alcohol use and found that cetuximab plus cisplatin/fluorouracil or carboplatin/fluorouracil improved the response rate (36% vs. 20%; *p* < 0.001) and median survival (10.1 vs. 7.4 months; *p* = 0.04) compared to the standard chemotherapy doublet of platinum/fluorouracil [47]. Another phase II trial (GORTEC 2014-01 TPExtreme) failed to demonstrate a survival benefit for the combination of platinum plus taxane and cetuximab versus the EXTREME regime [52]. At a median follow-up of 34 months, similar OS was observed between arms (14.5 vs. 13.4 months; *p* = 0.23), although there were less toxicity and delays in administration and more patients initiating cetuximab maintenance in the taxane plus platinum group.

Despite the availability of new combinations of cytotoxic chemotherapy with anti- EGFRs and the introduction of immunotherapy, the survival rates and prognosis of patients with advanced oral cancer remain unsatisfactory [53]. The development of chemoresistance greatly limits the effectiveness of treatment regimens, which is the reason why it is urgent to investigate and improve the understanding of its underlying mechanisms. Different mechanisms involving multiple pathways and/or processes—such as DNA repair, DNA damage response, drug transport, and apoptosis—contribute to resistance or sensitivity to cisplatin [53,54]. Among them, microRNAs seem to play a prominent role in determining resistance or sensitivity, through their action on molecules and/or pathways related to apoptosis, autophagy, hypoxia, cancer stem cells, NF-κB, and Notch1 [54,55]. In addition, the modulation of relevant microRNAs can effectively re-sensitize cancer cells to cisplatin regimens [54,55].

Other therapeutic options are under investigation, with targeted therapies holding promise for heavily pretreated SCCHN patients, traditionally with limited treatment alternatives. The use of farnesyltransferase inhibitors, such as tipifarnib, in patients with mutations in the HRAS proto-oncogene have shown encouraging results in a single-arm, open-label, phase II trial with ORR as primary endpoint. In total, 10 of the 22 patients enrolled had diagnosis of oral cavity primary tumors. The ORR with tipifarnib for evaluable patients was 55% (95% CI, 31.5–76.9%), and the median PFS was 5.6 months (95% CI, 3.6–16.4 months) versus 3.6 months (95% CI, 1.3–5.2 months) with the last prior therapy. The median OS was 15.4 months (95% CI, 7.0–29.7 months), and the most frequent treatment-emergent side effects were anemia (37%) and lymphopenia (13%) [56].

Cyclin-dependent kinase (CDK) 4- and 6-specific inhibitors, such as palbociclib, are also being explored in this setting. However, the results of the phase II PALATINUS trial were not auspicious, since the combination of cetuximab with palbociclib in HPV-negative, cetuximab-naïve patients failed to improve OS in platinum-resistant disease [57]. Conversely, the combination of palbociclib and cetuximab was shown to be active in platinum-resistant, cetuximab-resistant HPV-unrelated tumors in another phase II trial, raising the question of whether CDK inhibitors can be an option to revert cetuximab resistance [58].

Other molecules, such as the tyrosine kinase inhibitors (TKIs) afatinib and gefitinib, have also been explored in R/M SCCHN. Afatinib was compared to methotrexate in patients who progressed on or after platinum-based therapy in the phase III LUX-Head and Neck 1 trial [59]. Patients who received afatinib showed superior PFS than those who received methotrexate (2.6 vs. 1.7 months; *p* = 0.03), without significant OS differences between groups [59]. Subsequent biomarker analysis uncovered a subgroup of patients who may benefit more from this therapeutic approach: the group with p16-negative, EGFR-amplified, human epidermal growth factor receptor 3 (HER3)-low, and phosphatase and tensin homolog (PTEN)-high tumors [60]. Further studies are now required to validate these findings. Another randomized phase II trial compared afatinib and cetuximab in patients with R/M SCCHN who progressed on or after platinum-based therapy, showing comparable response rates with both agents [61]. Gefitinib also showed marginally improved overall response compared to methotrexate, specifically in recurrent head and neck tumors, but the addition of this TKI to docetaxel failed to improve outcomes in poor prognosis but otherwise unselected patients with metastatic head and neck cancer [62].

Other treatment strategies (Table 1) are currently being pursued, such as dual inhibition of the phosphatidylinositol-3-kinase/mammalian target of rapamycin (P13K/mTOR) pathway or the use of a pan-PI3K inhibitor in association with paclitaxel in NOTCH1-mutant meta- static head and neck carcinoma [63].

Future studies should focus on deepening the understanding of the process of carcinogenesis and exploring new therapeutic approaches focusing on aberrant methylation, synthetic lethal strategies for cancers with loss of tumor suppressor function, and downregulation of immunosuppressive signals in the tumor microenvironment (TME) [63].

Keypoints of Chemotherapy and Molecular Target Agents Section:The main treatments goals in patients with R/M SCCHN are to prolong survival and/or provide symptom palliation.Platinuam-based chemotherapy in combination with cetuximab is considered the standard of care for first line treatment in patients not suitable for immunotherapy.Cisplatin, taxanes, methotrexate, fluorouracil, or cetuximab can be used as a single-agent treatment.Doublet cytotoxic chemotherapy has been shown to increase the objective response rate compared with single-agent chemotherapy, although with no survival benefit.Based on the improvement in knowledge of SCCHN molecular biology, new compounds and approaches are being investigated for recurrent and metastatic setting.

## 4. Immunotherapy

A systematic analysis of the recently published Global Burden of Disease Study 1990–2017 measured the global quality of care of patients with SCCHN and concluded that, despite still being a neglected condition and often diagnosed in late stages, an effort has been made over the years to improve the management and care of these patients [2]. This has been mainly achieved due to advances in systemic treatment, namely the recent introduction of immunotherapy in the therapeutic armamentarium to fight the disease.

Immunotherapy has flourished in the last decade after the Nobel Prize-winning discovery of specific membrane proteins in the immune system and tumor cells. This discovery awarded Tasuku Honjo and James P. Allison the 2018 Nobel Prize of Physiology or Medicine for the discovery of the programmed death molecule-1 (PD-1) and cytotoxic T-lymphocyte antigen-4 (CTLA-4) on T-cells, respectively, which culminated in the development of the groundbreaking cancer therapy known as immune checkpoint blockade [64]. Tumor suppression of T-cell activation via PD-1 and/or CTLA-4 is a major escape mechanism of cancer cells, and the possibility of unlocking this suppression revolutionized the treatment paradigm of many types of cancer, including SCCHN.

Immunotherapy is currently a well-established player in the treatment armamentarium for recurrent/persistent and/or metastatic SCCHN, specifically the two PD-1-blocking monoclonal antibodies nivolumab and pembrolizumab, also known as checkpoint inhibitors. Nivolumab is approved for the treatment of platinum-resistant disease progressing within six months after definitive treatment with a platinum component or within first- line treatment with a platinum component [65], while pembrolizumab is indicated for platinum-sensitive disease progressing more than six months after definitive treatment with a platinum component and expressing PD-L1 [50].

PD-L1 expression in tumors and in lymphocytes and macrophages can be quantified through the combined positive score (CPS, defined by the number of PD-L1 staining cells (tumor cells) divided by the total number of viable tumor cells multiplied by 100) and is the current accompanying biomarker for the use of pembrolizumab in several solid tumors. Several other potential biomarkers are being explored for use in the clinical practice in the future.

The approval of nivolumab and pembrolizumab in R/M SCCHN was granted following the results of two pivotal phase III trials, Checkmate 141 (CM-141) and Keynote 3475-048 (KN-048), respectively, which showed the superiority of these agents over the standards of care at the time in this setting.

CM-141 was a randomized, open-label, phase III trial that assigned 361 patients with recurrent SCCHN whose disease progressed within 6 months after platinum-based chemotherapy in a 2:1 ratio to nivolumab every 2 weeks or standard, single-agent systemic therapy at the investigator’s choice (methotrexate, docetaxel, or cetuximab) [66]. The primary endpoint was OS. The study confirmed a significantly longer OS with nivolumab compared to standard therapy (hazard ratio (HR) for death, 0.70; 97.73% CI, 0.51–0.96; *p* = 0.01), with a 1-year survival estimate approximately 19% longer with the anti-PD-1 compared to standard therapy in the overall intent-to-treat (ITT) population (36.0% vs. 16.6%) [66]. With a minimum follow-up of 11.4 months, 7% of patients in the nivolumab arm and 1% of patients in the investigator’s choice arm were still on treatment in the ITT population, with nivolumab continuing to improve the OS compared to standard therapy. The 18-month OS rate nearly tripled with nivolumab (21.5% vs. 8.3%) and was consistent among subgroups.

The phase III KN-048 was a randomized trial of patients with untreated locally incurable R/M SCCHN who were stratified by PD-L1 expression, p16 status, and PS and randomly allocated in a 1:1:1 ration to pembrolizumab alone, pembrolizumab plus a platinum agent and 5-fluorouracil (pembrolizumab with chemotherapy), or cetuximab plus a platinum agent and 5-fluorouracil (cetuximab with chemotherapy, the standard first-line therapy at that time) [50]. The study had OS and PFS in the ITT population as coprimary endpoints. KN-048 was a complex study with 14 primary hypotheses: superiority of pembrolizumab alone and of pembrolizumab with chemotherapy versus cetuximab with chemotherapy for OS and PFS in the PD-L1 CPS ≥20 and CPS ≥1 and overall study populations, and non-inferiority of pembrolizumab alone and of pembrolizumab with chemotherapy versus cetuximab with chemotherapy for OS in the overall study population. At the second interim analysis, pembrolizumab alone improved the OS versus cetuximab with chemotherapy in the CPS ≥ 20 (median 14.9 vs. 10.7 months, HR 0.61, *p* = 0.0007) and CPS ≥ 1 (12.3 vs. 10.3, HR 0.78, p 0.0086) populations, and was non-inferior to the standard regimen in the overall study population (median c vs. 10.7 months, HR 0.85, *p* = 0.0456). Pembrolizumab with chemotherapy improved the OS versus cetuximab with chemotherapy in the overall study population (13.0 vs. 10.7 months, HR 0.77, p 0.0034) at the second interim analysis and in the CPS ≥ 20 (14.7 vs. 11.0, HR 0.60, p 0.0004) and CPS ≥ 1 (13.6 vs. 10.4, HR 0.65, *p* < 20) subgroups, not having been presented separately for the absence of comparison between the pembrolizumab monotherapy and pembrolizumab plus chemotherapy groups. The study authors suggested that pembrolizumab in monotherapy could be a good option for patients with low symptom burden, while pembrolizumab with chemotherapy could be more suitable for more symptomatic patients in need for a rapid objective response, for patients with low PD-L1 expression, and for patients with local disease recurrence only.

Data on patient-reported QoL are available for the two drugs. This was a secondary study endpoint in the CM-141 trial, which showed that nivolumab provided improved patient wellbeing [66]. In this trial, patient-reported QoL measures were similar at baseline for the two study drugs, but deteriorated as treatment progressed for patients in the standard therapy arm, who reported clinically meaningful worsening of physical, role, and social functioning, as assessed by the European Organization for Research and Treatment of Cancer 30-Item QoL Questionnaire (EORTC QLQ-C30), and of pain and sensory and social-contact problems, as assessed by its head and neck module QLQ-H&N35 [66]. Conversely, these QoL measures remained stable or even slightly improved for patients treated with nivolumab. QoL data of pembrolizumab were reported in the KN-040 trial, a study prior to the KN-048 trial and with a similar design to CM-141 [67]. In the trial, EORTC QLQ-C30 global health status and QoL scores remained stable compared to the baseline in patients treated with the anti-PD-1. HRQoL compliance at week 15 was 75.3% with pembrolizumab and 74.6% with investigator’s choice therapy, and the median time to deterioration of global health status and QoL measures was 4.8 and 2.8 months, respectively (HR 0.79, 95% CI 0.59–1.05). Oral cavity tumors accounted for 45% and 55.4% of tumors in the nivolumab and standard therapy arms in the CM-141 trial, respectively, and for 27–30% of tumors in each of the three arms of the KN-048 trial.

Other immune checkpoint inhibitors have also been evaluated both in the platinum-sensitive and in the platinum-resistant setting with negative results. The phase III KESTREL trial was randomized as 2:1:1 to durvalumab alone, durvalumab plus tremelimumab, and the EXTREME regimen. The primary endpoint was OS for durvalumab monotherapy vs. EXTREME in PD-L1 high expressers (tumor cell expression of >50% or tumor-infiltrating lymphocyte expression >25%), and the secondary endpoint of OS was for durvalumab plus tremelimumab vs. EXTREME for all patients. The trial failed to meet these endpoints [68]. Another approach accessed was the combination of Ipilimumab and Nivolumab in the CHECKMATE 651 phase III trial that randomly assigned 1:1 to nivolumab plus ipilimumab or EXTREME. The primary endpoints were overall survival (OS) in the all randomly assigned and programmed death-ligand 1-combined positive score (CPS) ≥ 20 populations. The trial did not meet its primary endpoints of OS in all randomly assigned or CPS ≥ 20 populations [69]. In the platinum-resistant setting, the phase III EAGLE trial randomized platinum failure R/M SCCHN patients to durvalumab plus tremelimumab, durvalumab monotherapy, or investigator choice standard of care chemotherapy. This trial was dually powered for OS comparison of durvalumab and combination durvalumab plus tremelimumab separately, compared to chemotherapy. There was no difference in OS with durvalumab (HR 0.88, 95% CI [0.72, 1.08], *p* = 0.20) or durvalumab plus tremelimumab. (HR 1.04, 95% CI [0.85, 1.26], *p* = 0.76) compared to chemotherapy [70]. Accepting the limitations of cross-trial comparisons, it is notable that while the median OS with durvalumab was similar to nivolumab in Checkmate 141 (7.6 vs. 7.5 months, respectively), the median OS of the control arm was numerically longer in EAGLE compared to CHECKMATE 141 (8.3 months vs. 5.1 months respectively). Exploratory analysis from EAGLE suggests that this higher than expected OS in the control group may have come from imbalance in baseline characteristics (higher percentage of ECOG PS 0 and distant metastasis only in the control arm); increased usage of paclitaxel in the control arm, which was not a choice in CHECKMATE 141 or KEYNOTE 040; and subsequent receipt of anti-PD-1 therapy [68]. Overall, the OS results of the two immune checkpoint inhibitors (pembrolizumab and nivolumab) were groundbreaking in this late stage of SCCHN. However, only a few patients benefited from the long survival times achieved, and even those patients eventually experienced disease progression. This raises several questions, namely what patient features made them respond differently to treatment, what are the prognostic and predictive factors in the disease, and how to treat patients after immunotherapy, among others. It is acknowledged that head and neck cancer is characterized by a heterogeneous immune landscape that impacts treatment response, which includes gene mutations, amplifications, fusions, and copy number alterations [71]. The current consensus of molecular classification of SCCHN groups these tumors into classical, basal, mesenchymal, and atypical subtypes, each with unique gene expression profiles and biological characteristics [71]. These subtypes are represented in all anatomic sites and clinical stages, except for hypopharyngeal cancers, which lack the basal subtype [71]. However, and despite its predictable benefits, this molecular classification is not yet applied in the routine clinical practice to individualize treatment.

In a recent study in a mouse model of oral cavity cancer with high concordance with human oral cavity cancer, a high rate of unique tumor neoantigens was associated with response to immunotherapy [72]. The study also showed that tumors with higher neoantigen load, thereby being responsive to immunotherapy, had a more profound antigen-specific lymphocyte response than those with lower neoantigen load. In the same study, the analysis of RNA sequencing data from a cohort of SCCHN from The Cancer Genome Atlas tumor bank revealed that both HPV-negative and -positive tumors with strong immune infiltrate of CD8+ tumor-infiltrating lymphocytes (TILs) and natural killer cells had a survival benefit compared to tumors with less robust immune infiltrate [72]. This suggests that TILs may be an important marker in the future for SCCHNs.

The TME seems to play a relevant role in the response to immunotherapy, with pre-clinical research currently ongoing. The immune landscape of HPV-positive SCCHN seems to have an inflammatory, yet immunosuppressed TME, with heavy immune infiltrates of CD8+PD-1+ T-cells and regulatory T-cells [71]. This suggests that HPV-positive tumors are more prone to respond to immune activation stimuli when the existing immune-suppressive elements are eliminated or blocked. On the other hand, the TME of HPV-negative SCCHN is highly immunosuppressed and characterized by low levels of immune infiltrates, making these tumors potentially poor responders to immunotherapy, including checkpoint inhibitors [71]. It is yet to be determined whether HPV-positive tumors are present in other locations besides the oropharynx.

Several immune response escape mechanisms have been described in SCCHN, from secretion of soluble factors, such as immunosuppressive cytokines and metabolites, to depletion of local micronutrients and coaptation of checkpoint pathways [72]. The development of strategies to overcome this immunosuppression, eventually resorting to antibodies and vaccines, will be a course to take in the future. Oncolytic viral therapy, vaccine therapies, and adoptive cell transfer (i.e., tumor-specific T-cells that are expanded ex vivo and returned to the patient to kill tumor cells, theoretically generating long-lasting memory against recurrence) are promising fields to explore in the treatment of SCCHN, more likely in combination regimens than in monotherapy [72].

The optimal therapeutic sequencing after immunotherapy is yet to be determined. In patients still fit for treatment, any of the previous therapies approved for SCCHN has shown very poor response rates, albeit with a trend towards slightly better outcomes than without prior immunotherapy. It has been reported that early exposure to immune check-point blockade might induce durable alterations in the TME regardless of treatment response, thereby modifying tumor sensitivity to subsequent therapies [73]. Overall, although the treatment outcomes in R/M SCCHN evolved substantially with the introduction of immunotherapy, the optimal therapy sequencing will allow us to maximally prolong survival while maintaining the best QoL remains elusive.

Several trials are ongoing with immunotherapy combinations and combinations of immunotherapy and targeted therapies, which will hopefully uncover new and improved strategies for the treatment of R/M SCCHN in general and oral cavity cancer in particular.

The association of afatinib and pembrolizumab is one approach being study, in a single-arm phase II trial with the primary endpoint of ORR. From January of 2019 to March 2020, the study enrolled 29 eligible patients with the primary results suggesting that this association may improve response in this subset of patients [74]. Another option is the combination of pembrolizumab and lenvatinib, in the phase III LEAP-010 trial where the patients with R/M SCCHN CPS ≥ 1 will be randomized to pembrolizumab plus pembrolizumab or pembroluzumab plus placebo. This trial is currently recruiting and is supported byinitial data, where the association achieved objectives responses in both heavily pretreated and anti-PD1 refractory R/M SCCHN [75].

Even more promising is the neoadjuvant treatment of early stages of the disease, namely with immune checkpoint inhibitors in monotherapy or combination, immune checkpoint inhibitors with other immunotherapy agents, and other drug combinations, such as TKIs, which will hopefully improve the early control of the disease and hence the survival of these patients. Several trials are underway after promising evidence of downstaging and even pathologic complete responses in phase I and II trials. (Table 2) An example is the SNOW window-of-opportunity study, designed to investigate the immune and molecular effects of preoperative sitravatinib (a TKI targeting TYRO3, AXL, MERTK, and the VEGF receptor family) and nivolumab in patients with SCCHN [76]. These and other results are awaited, with the expectation that they may contribute to changing the course of this still dramatic disease.

Keypoints of Immunotherapy Section:Immune checkpoint inhibitors changed the treatment landscape of head and neck cancer. Immune checkpoint inhibitors can be used not only in the platinum-resistant setting of R/M SCCHN (Checkmate 141), but also in the platinum-sensitive setting based on results from KEYNOTE-048, where pembrolizumab was approved in the first line setting for R/M SCCHN in monotherapy or in combination with chemotherapy according to CPS determination.Novel combinations of immunotherapies or target agents are under investigation.Predictive biomarkers remain a challenge to improve patient selection.

## 5. Conclusions

The treatment paradigm of R/M SCCHN is currently undergoing an unprecedented evolution. The oral cavity topography includes a remarkable number of tightly packed noble structures, such as blood vessels and pivotal nerves. These structural constraints pose unique challenges to the multidisciplinary team in the management of the disease. While surgeons are usually unable to perform salvage resections with good functional results, radiation oncologists seek to balance the radiation dose capable of achieving ap- propriate local control with minimal collateral damage to neighboring vital organs with available and emerging radiotherapy techniques (e.g., IMRT, SBRT, IMPT, and brachytherapy), and medical oncologists aim for new targets for immunomodulatory and targeted therapies, as the intensification limits of conventional chemotherapy have been largely met.

Immunotherapy has shown promising survival outcomes in R/M SCCHN, both as monotherapy and in combination with chemotherapy, with tolerable toxicity and improvements in patient-reported outcomes. The pool of immunotherapy-eligible patients will predictably expand as more data become available. Still, this therapeutic modality is not suitable for all patients. The rate of patients who respond to immunotherapy is below 20%, and about 60% of patients who receive immune checkpoint inhibitors experience immune-related adverse effects. In addition, the ORR in patients treated with immunotherapy is far from satisfactory, which may be a significant issue, particularly in patients affected by very symptomatic neck masses [50,65]. Therefore, although it is unquestionable that immunotherapy is changing the treatment landscape of R/M SCCHN, data are still lacking to optimize its use in this indication.

Several unmet needs persist, namely regarding predictive biomarkers, better knowledge of the process of carcinogenesis, and optimal sequencing of available therapies that allow us to maximally prolong patient survival while maintaining the highest possible QoL.

Although reirradiation is the local alternative to surgery for unresectable disease, its use in R/M SCCHN poses the dilemma to radiation oncologists of salvage/curative intent using high-dose radiation versus palliative treatment. One must remember that not all patients submitted to high-dose radiotherapy achieve 2-year OS, and some may suffer from significant cumulative toxicity, including the highly worrying carotid artery blow-out. Based on the limited evidence available, the American Radium Society (ARS) Committee defined that radiotherapy with curative intent (fractionated reirradiation with 60–70 Gy with IMRT) with concurrent systemic therapy should be offered to patients for whom a PFS of at least 2 years is expected, accepting a limited risk of grade 5 toxicity (8% reported in studies) [75]. The use of elective lymph node irradiation is currently not considered appropriate. The plethora of prescription doses used in palliative radiotherapy is wide, and its selection depends on several factors. Some doses can be safely combined with systemic therapies in a sequential way. However, the use of reirradiation in certain organs at risk (e.g., spinal cord and brainstem) is limited by the previously accumulated dose.

Clinical trials are ongoing to assess the benefit of adding radiotherapy in patients submitted to immunotherapy, with some investigators arguing that research should also include patients with metastases, a population excluded from most trials.

In conclusion, the management of R/M SCCHN remains challenging. Despite the recent emergence of immunotherapy and progresses in radiation modalities, it is crucial to improve the knowledge of the process of carcinogenesis and explore new therapeutic approaches. The continuous development of specific targeted and immune therapies, together with the optimization of radiation therapies, will hopefully pave the way for better patient outcomes and QoL.

## Figures and Tables

**Table 1 diagnostics-13-00099-t001:** Clinical trials of chemotherapy and molecular-targeted agents in R/M SCCHN currently recruiting patients (ClinicalTrials.gov, as of 6 August 2022).

Phase	Trial Identification
Phase I	Cabozantinib in Combination With Cetuximab in Patients With Recurrent or Metastatic Head and Neck Squamous Cell Cancer *ClinicalTrials.gov Identifier: NCT03667482*
Phase I/II	Combination Trial of Tipifarnib and Alpelisib in Adult Recurrent/Metastatic Head and Neck Squamous Cell Carcinoma (R/M HNSCC) *ClinicalTrials.gov Identifier: NCT04997902*
Phase II	Duvelisib Plus Docetaxel in Recurrent/Metastatic HNSC *ClinicalTrials.gov Identifier: NCT05057247*
Phase II	Second-Line Chemotherapy Combined With Endostatin for Recurrent/Metastatic HN Epithelial Tumors (SLICER) *ClinicalTrials.gov Identifier: NCT03989830*
Phase II	Paclitaxel Plus Cetuximab After First-line Checkpoint Inhibitor Failure *ClinicalTrials.gov Identifier: NCT04278092*
Phase II	Cetuximab After Immunotherapy for the Treatment of Head and Neck Squamous Cell Cancer *ClinicalTrials.gov Identifier: NCT04375384*

**Table 2 diagnostics-13-00099-t002:** Clinical trials of immunotherapy agents in R/M SCCHN that are currently recruiting and not yet recruiting patients (ClinicalTrials.gov, as of 6 August 2022).

Phase	Trials Recruiting Patients	Trials Not Yet Recruiting Patients
Phase Ib		Open-Label, Single-Arm Dose-Expansion Study of IK-175, an Oral Aryl Hydrocarbon Receptor Inhibitor, in Combination With Nivolumab in Patients With Primary PD-1 Inhibitor Resistant Metastatic or Locally Incurable, Recurrent HNSCC *ClinicalTrials.gov Identifier: NCT05472506*
Phase II	Study of Ipatasertib in Combination With Pembrolizumab for First-Line Treatment of Recurrent or Metastatic Squamous Cell Cancer of the Head and Neck. *ClinicalTrials.gov Identifier: NCT05172258*	(BiCaZO) Study of Combining Cabozantinib and Nivolumab in Patients With Advanced Solid Tumors (IO Refractory Melanoma or HNSCC) Stratified by Tumor Biomarkers—an immunoMATCH Pilot Study *ClinicalTrials.gov Identifier: NCT05136196*
Phase II	Study of First-Line Weekly Chemo/Immunotherapy for Metastatic Head/Neck Squamous Cell Carcinoma Patients *ClinicalTrials.gov Identifier: NCT04858269*	Multicenter, Open-Label, Single-Arm Study to Evaluate the Safety and Efficacy of Oral NRC-2694-A in Combination With Paclitaxel in Patients With Recurrent and/or Metastatic Head and Neck Squamous Cell Carcinoma, Who Progressed on or After Immune Checkpoint Inhibitor Therapy *ClinicalTrials.gov Identifier: NCT05283226*
Phase II	Trial of Pembrolizumab and Cabozantinib in Patients With RM SCCHN *ClinicalTrials.gov Identifier: NCT03468218*	
Phase II	Study of Tadalafil and Pembrolizumab in Recurrent or Metastatic Head and Neck Squamous Cell Carcinoma *ClinicalTrials.gov Identifier: NCT03993353*	
Phase II	Trial of the Efficacy and Safety of the Combination of Cemiplimab and Low-Dose Paclitaxel and Carboplatin in Patients With Recurrent/Metastatic Squamous Cell Carcinoma of the Head and Neck *ClinicalTrials.gov Identifier: NCT04862650*	
Phase II	Prospective, Single-Center, Randomized, Controlled Study of TC (Docetaxel and Carboplatin) Regimen With or Without Nimotuzumab in Recurrent Metastatic Oral Squamous Cell Carcinoma *ClinicalTrials.gov Identifier: NCT04367909*	
Phase II		A Study of Pembrolizumab in Combination With Chemotherapy for Head and Neck Cancer *ClinicalTrials.gov Identifier: NCT05420948*
Phase II/III	Trial of Chemotherapy + Cetuximab vs Chemotherapy + Bevacizumab vs Atezolizumab + Bevacizumab Following Progression on Immune Checkpoint Inhibition in Recurrent/Metastatic Head and Neck Cancers *ClinicalTrials.gov Identifier: NCT05063552*	
Phase III	Randomized, Double-Blind, Multicenter, Global Study of Monalizumab or Placebo in Combination With Cetuximab in Participants With Recurrent or Metastatic Squamous Cell Carcinoma of the Head and Neck Previously Treated With an Immune Checkpoint Inhibitor *ClinicalTrials.gov Identifier: NCT04590963*	
Phase II	KEYSTROKE/RTOG 3507 *ClinicalTrials.gov Identifier: NCT03546582*	
Phase II	EA3191 *ClinicalTrials.gov Identifier: NCT04671667*	
Phase II	Reirradiation With NBTXR3 in Combination With Pembrolizumab for the Treatment of Inoperable Locoregional Recurrent Head and Neck Squamous Cell Cancer *ClinicalTrials.gov Identifier: NCT 04834349*	

## Data Availability

Not applicable.

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
