# Peer review of "Radiotherapy, Chemotherapy and Immunotherapy—Current Practice and Future Perspectives for Recurrent/Metastatic Oral Cavity Squamous Cell Carcinoma"

_diagnostics, 2022, doi:10.3390/diagnostics13010099_

Round 1
Reviewer 1 Report
Dear Authors,
Thank you for your submission
The authors conducted a comprehensive literature review regarding therapeutic options for recurrent/metastatic head and neck cancers.
The topic is interesting and the manuscript has been well-written and discussed.
Minor comments:
1. Although at the end of the radiotherapy section, the authors have mentioned the potential role of a combination of radiotherapy and immunotherapy; however, due to the paucity of evidence in the literature and considering the attractiveness and novelty of this topic, I recommend that they add one more paragraph in this regard to the article.
2. The amount of evidence presented in the paper is so much, therefore, adding the key points to the end of each section would be more efficient and attractive for the readers.
Round 1
Reviewer 1 Report:
Dear Authors,
Thank you for your submission
The authors conducted a comprehensive literature review regarding therapeutic options for recurrent/metastatic head and neck cancers.
The topic is interesting and the manuscript has been well-written and discussed.
Minor comments:
- Although at the end of the radiotherapy section, the authors have mentioned the potential role of a combination of radiotherapy and immunotherapy; however, due to the paucity of evidence in the literature and considering the attractiveness and novelty of this topic, I recommend that they add one more paragraph in this regard to the article.
- The amount of evidence presented in the paper is so much, therefore, adding the key points to the end of each section would be more efficient and attractive for the readers.
Author Response:
Point 1: We add the following information regarding the potential role of combination of radiotherapy and immunotherapy.
Some currently ongoing trials may uncover new directions for the treatment of R/MSCCHN, as the combination of radiotherapy with immunotherapy which is currently a hot topic for investigation. The pillar concept for this combination is a synergistic effect, by which neoantigens produced during radiotherapy treatment and the immunotherapy agents (ie: immune checkpoint inhibitors) promote immunological synapses in order to intensify the host immune system against the cancer cells. This can happen near the area of irradiation but also over distant metastasis (known as abscopal effect). This combination can be useful for locally advanced disease and metastatic disease, especially if limited oligoprogression is observed while on isolated immunotherapy, promoting the so-called “turning cold tumor to hot tumor” effect. Although more established for the combination of immune therapies, the immune effect produced by radiotherapy can also play a role in this setting.
The concept is feasible but there are still impeding questions beyond the scope of the paper that are under investigation: What are the best immunotherapy agents for the combination? Which is the ideal biomarker(s)? Ideal timings for the introduction of each therapy?​ What are the appropriate RT technique, prescription dose and treatment volumes of interest to enhance the immunological effect?​
Ongoing trials without results for a plethora of cancer diseases and settings are being conducted with very few SCCHN cancers, especially for oral cavity carcinomas.
Also add as reference the following paper:
[35] Zhang J, Huang D, Saw PE, Song E. Turning cold tumors hot: from molecular mechanisms to clinical applications. Trends Immunol. 2022 Jul;43(7):523-545. doi: 10.1016/j.it.2022.04.010. Epub 2022 May 25. PMID: 35624021.
Point 2: In the end of each section, we add some keypoints to be more attractive to readers.
Reviewer 2 Report
Dear editor:
The authors conducted a systemic review regarding the chemotherapy, radiotherapy and immunotherapy for patients with recurrent or metastatic head and neck cancers. This review is very comprehensive and would be helpful for physician who treated patients with recurrent or metastatic head and neck cancer. However, there are some issues needed for discussion.
1. In the part of radiotherapy, the authors discussed the adjuvant role for locoregional recurrent head and neck cancer after curative surgery. Given that the title of this manuscript is “recurrent or metastatic head and neck cancer” which means unresectable disease, the authors should review more regarding the role of palliative radiotherapy for these patients. For example, the prognostic impact of re-irradiation of the overlapping area or palliative radiotherapy of metastatic lesions are worth for discussion.
2. In the part of immunotherapy, the phase III studies with anti-CTLA4, ipilimumab or tremelimumab, for recurrent or metastatic head and neck cancer patients are all negative. Maybe, the authors should also mention about these important data.
3. For immunotherapy combination, there are some published phase II study with positive results. For example, pembrolizumab plus afatinib or pembrolizumab plus lenvatinib are the most famous studies. The authors should also have some discussion about these two studies.
Round 1
Reviewer 2 Report:
Dear editor:
The authors conducted a systemic review regarding the chemotherapy, radiotherapy and immunotherapy for patients with recurrent or metastatic head and neck cancers. This review is very comprehensive and would be helpful for physician who treated patients with recurrent or metastatic head and neck cancer. However, there are some issues needed for discussion.
- In the part of radiotherapy, the authors discussed the adjuvant role for locoregional recurrent head and neck cancer after curative surgery. Given that the title of this manuscript is “recurrent or metastatic head and neck cancer” which means unresectable disease, the authors should review more regarding the role of palliative radiotherapy for these patients. For example, the prognostic impact of re-irradiation of the overlapping area or palliative radiotherapy of metastatic lesions are worth for discussion.
- In the part of immunotherapy, the phase III studies with anti-CTLA4, ipilimumab or tremelimumab, for recurrent or metastatic head and neck cancer patients are all negative. Maybe, the authors should also mention about these important data.
- For immunotherapy combination, there are some published phase II study with positive results. For example, pembrolizumab plus afatinib or pembrolizumab plus lenvatinib are the most famous studies. The authors should also have some discussion about these two studies.
Author Response:
Point 1: We add some changes in the radiotherapy section with more focus in palliative protocols in this subset of patients.
In patients with low estimated survival (up to 4 months) unfit for other cancer treatments, the main goal is comfort and symptom relief. Ideally, the duration of the complete radiotherapy scheme should be no longer than two weeks. Possible schemes include QUAD-SHOT, 20 Gy in 5 fractions (1 fraction per day), and 28 Gy in 3 fractions (in days 0, 7 and 21). [30-33] For patients with an estimated survival between 4 and 12 months, QUAD SHOT with or without chemotherapy is an option, as well as 20 Gy in 5 fractions (4 Gy daily), 30 Gy in 5 fractions twice per week, or 40 Gy in 10 fractions twice per week. [30-33] For patients with an estimated survival over 12 months, more aggressive treatment can be considered. For patients with no indication for further treatments, but nevertheless in a good clinical condition, a hypofractionated regimen with 50 Gy in 16 fractions (3.125 Gy/day) or 52.5Gy in 15 fractions (3.5 Gy/day) is recommended. [30]. The fact that palliative radiation has a primary goal of symptom relief should not hinder the fact that it can also contribute to local control and even survival. The QUAD-SHOT study reported over 50 % of objective responses, with a median OS of 5.7 months, which is promising given that those were patients not amenable to curative therapy. “Christie scheme” (3.125 Gy per fraction) reported an OS of 40% at 1 year and a median survival time of 17 months. Even the more modest “0-7-21” regimen showed a median 6-month OS of 51 % with a 39 % PFS within the irradiated volume.
Also add as reference the following paper:
[31] Nguyen NT, Doerwald-Munoz L, Zhang H, Kim DH, Sagar S, Wright JR, Hodson DI. 0-7-21 hypofractionated palliative radiotherapy: an effective treatment for advanced head and neck cancers. Br J Radiol. 2015 May;88(1049):20140646. doi: 10.1259/bjr.20140646. Epub 2015 Feb 19. PMID: 25694259; PMCID: PMC4628471.
Point 2: We add the results of trials as KESTREL, EAGLE and CHECKMATE 651 as is possible to see in this paragraph added to the paper:
Other immune checkpoint inhibitors have also been evaluated both in the platinum sensitive and in the platinum resistant setting with negative results. The phase III KESTREL trial randomized 2:1:1 to durvalumab alone, durvalumab plus tremelimumab, and the EXTREME regimen. The primary endpoint was OS for durvalumab monotherapy vs. EXTREME in PD-L1 high expressers (tumor cell expression of >50% or tumor-infiltrating lymphocyte expression >25%) and secondary endpoint of OS for durvalumab plus tremelimumab vs. EXTREME for all patients. The trial failed to meet these endpoints. [68] Other approach accessed was the combination of Ipilimumab and Nivolumab in the CHECKMATE 651 phase III trial that randomly assigned 1:1 to nivolumab plus ipilimumab or EXTREME. Primary end points were overall survival (OS) in the all randomly assigned and programmed death-ligand 1 combined positive score (CPS) ≥ 20 populations. The trial did not meet its primary end points of OS in all randomly assigned or CPS ≥ 20 populations. [69] In the platinum resistant setting, the phase III EAGLE trial randomized platinum failure R/M SCCHN patients to durvalumab plus tremelimumab, durvalumab monotherapy, or investigator choice standard of care chemotherapy. This trial was dually powered for OS comparison of durvalumab and combination durvalumab plus tremelimumab separately, compared to chemotherapy. There was no difference in OS with durvalumab (HR 0.88, 95% CI [0.72, 1.08], p=0.20) or durvalumab plus tremelimumab. (HR 1.04, 95% CI [0.85, 1.26], p=0.76) compared to chemotherapy. [70] Accepting the limitations of cross-trial comparisons, it is notable that while the median OS with durvalumab was similar to nivolumab in Checkmate 141 (7.6 vs. 7.5 months, respectively), the median OS of the control arm was numerically longer in EAGLE compared to CHECKMATE 141 (8.3 months vs. 5.1 months respectively). Exploratory analysis from EAGLE suggests that this higher-than expected OS in the control group may have come from imbalance in baseline characteristics (higher percentage of ECOG PS 0 and distant metastasis only in the control arm), increased usage of paclitaxel in the control arm, which was not a choice in CHECKMATE 141 or KEYNOTE 040, and subsequent receipt of anti-PD-1 therapy. [68]
Also add as reference the following papers:
[68] Hsieh RW, Borson S, Tsagianni A, Zandberg DP. Immunotherapy in Recurrent/Metastatic Squamous Cell Carcinoma of the Head and Neck. Front Oncol. 2021 Sep 1;11:705614.
[69] Haddad RI, Harrington K, Tahara M, et al. A. Nivolumab Plus Ipilimumab Versus EXTREME Regimen as First-Line Treatment for Recurrent/Metastatic Squamous Cell Carcinoma of the Head and Neck: The Final Results of CheckMate 651. J Clin Oncol. 2022 Dec 6:JCO2200332.
[70] Ferris RL, Haddad R, Even C, et al. L. Durvalumab with or without tremelimumab in patients with recurrent or metastatic head and neck squamous cell carcinoma: EAGLE, a randomized, open-label phase III study. Ann Oncol. 2020 Jul;31(7):942-950.
Point 3: We add the results of trials where the combination of pembrolizumab plus afatinib or pembrolizumab plus Lenvatinib has been studied in order to show new treatment possibilities in the future
The association of afatinib and pembrolizumab, is one of approach being study, in a single arm phase II trial with the primary endpoint of ORR. From January of 2019 to March 2020, the study enrolled 29 eligible patients with the primary results suggesting that this association may improve response in this subset of patients. [74] Other option is the combination of pembrolizumab and lenvatinib, in the phase III LEAP-010 trial where the patients with R/M SCCHN CPS ≥1 will be randomized to pembrolizumab plus pembrolizumab or pembroluzumab plus placebo. This trial is currently recruiting, and is supported by the initial data where the association achieved objectives responses in both heavily pretreated and anti-PD1 refractory R/M SCCHN [75]
Also add as reference the following papers:
[74] Kao HF, Liao BC, Huang YL, et al. Afatinib and Pembrolizumab for Recurrent or Metastatic Head and Neck Squamous Cell Carcinoma (ALPHA Study): A Phase II Study with Biomarker Analysis. Clin Cancer Res. 2022 Apr 14;28(8):1560-1571.
[75] Chen TH, Chang PM, Yang MH. Combination of pembrolizumab and lenvatinib is a potential treatment option for heavily pretreated recurrent and metastatic head and neck cancer. J Chin Med Assoc. 2021 Apr 1;84(4):361-367.